# Adherence to Diet and Meal Timing in a Randomized Controlled Feeding Study of Time-Restricted Feeding

**DOI:** 10.3390/nu14112283

**Published:** 2022-05-29

**Authors:** Beiwen Wu, Karen White, May Thu Thu Maw, Jeanne Charleston, Di Zhao, Eliseo Guallar, Lawrence J. Appel, Jeanne M. Clark, Nisa M. Maruthur, Scott J. Pilla

**Affiliations:** 1Division of General Internal Medicine, Johns Hopkins University School of Medicine, Baltimore, MD 21287, USA; kwhite33@jhmi.edu (K.W.); mmaw1@jhmi.edu (M.T.T.M.); eguallar@jhu.edu (E.G.); lappel@jhmi.edu (L.J.A.); jmclark@jhmi.edu (J.M.C.); maruthur@jhmi.edu (N.M.M.); spilla1@jhmi.edu (S.J.P.); 2Division of Epidemiology, Dalla Lana School of Public Health, University of Toronto, Toronto, ON M5S 1A1, Canada; 3Prosserman Centre for Population Health Research, Lunenfeld-Tanenbaum Research Institute, Sinai Health, Toronto, ON M5G 1X5, Canada; 4Department of Epidemiology, Johns Hopkins University Bloomberg School of Public Health, Baltimore, MD 21287, USA; jeannec@jhmi.edu (J.C.); dizhao@jhu.edu (D.Z.); 5Welch Center for Prevention, Epidemiology and Clinical Research, Johns Hopkins University, Baltimore, MD 21287, USA; 6Department of Health, Behavior & Society, Johns Hopkins University Bloomberg School of Public Health, Baltimore, MD 21287, USA

**Keywords:** time-restricted feeding, controlled feeding study, nutrition interventions, dietary adherence, compliance, diet monitoring

## Abstract

Adherence is critical in feeding studies to determine the efficacy of dietary interventions. This time-restricted intake of meals (TRIM) investigation was a controlled feeding study that randomized 41 participants to follow 12 weeks of time-restricted feeding (TRF) or a usual feeding pattern (UFP). Adherence was optimized through careful screening and participant orientation, flexibility in beverages and seasonings, and frequent contact between participants and staff. Adherence was measured daily using a self-administered diary form. We calculated the percentage of participant-days with perfect adherence to meal timing (ate all meals within their designated time window) and to food consumption (ate all study food and no non-study food). Adherence was compared between study arms, days of the week, and weeks of the study period using generalized estimating equations (GEE) regression. There was perfect adherence to meal timing on 87% of participant-days and to food consumption on 94% of participant-days, with no significant difference by arm. In UFP, but not TRF, participants had lower adherence to meal timing over the weekend (*p*-value = 0.002) and during the first two weeks of intervention (*p*-value = 0.03). A controlled feeding study randomizing free-living individuals to different meal timings achieved a high degree of adherence to meal timing and food consumption, utilizing multiple strategies.

## 1. Introduction

In efficacy studies, adherence is critical in nutrition research to understand the true effect of dietary interventions on physiological responses and health outcomes. In a controlled feeding study, participants are asked to consume all foods and beverages prepared by the research kitchen and no foods from outside the study [1,2,3,4,5,6,7,8,9,10,11,12]. When controlled feeding studies are conducted among free-living participants, achieving dietary adherence may be challenging, especially for studies with a shorter menu cycle, a longer feeding period, restricted diets, or greater participant time commitment [1,2,13]. 

Adherence may be especially challenging in studies where nutrient targets are allocated to specific time frames, including studies of time-restricted feeding (TRF), i.e., where food is consumed only during a specific portion of the day. In a typical feeding study, participants may choose when to consume their meals as long as all food is consumed within a given day; a study of time-restricted feeding cannot offer that flexibility. There are limited data on how to measure and optimize adherence in studies of time-restricted feeding among free-living individuals [6,11,14,15,16].

Thus, we analyzed adherence in the time-restricted intake of meals (TRIM) study, which was a randomized controlled feeding study that examined the effects of TRF on weight and metabolic factors. In this study, we describe our approach to maximizing and assessing adherence and examine adherence to the study diet and timing of meals.

## 2. Materials and Methods

### 2.1. Study Design

TRIM was a randomized controlled feeding study conducted at the Johns Hopkins ProHealth Clinical Research Unit in West Baltimore, MD, USA, to test the effects of TRF as compared with a usual feeding pattern (UFP) on weight (primary outcome) and other metabolic outcomes. The study was approved by the Johns Hopkins University School of Medicine Institutional Review Board. TRIM included adults aged 21 to 69 years with body mass index (BMI) from 30 to 50 kg/m^2^ and with prediabetes or well-controlled type 2 diabetes (HbA1c from 5.7 to 6.9%) not using glucose-lowering medications. Exclusion criteria included moderate to severe sleep apnea; renal dysfunction with glomerular filtration rate <30 mL/min/1.73 m^2^; shift work, sleep or circadian rhythm disorders; use of unstable doses of hypertension medications; use of medications that affect weight or sleep; drinking >1 alcoholic beverage per day; active substance use or significant psychiatric disorder; pregnancy, breastfeeding, a plan to become pregnant; routinely eating within a compressed time window within the past year; weight loss or gain of ≥5% during the past 6 months; prior bariatric surgery; or other conditions that could interfere with study participation. Participants were randomized 1:1 to two parallel study arms receiving the same diet but with different meal timing throughout the day (Table A1). The TRF arm consumed all food between 08:00 and 18:00 with 80% of calories before 13:00, and the UFP arm consumed food between 08:00 and 24:00 with 50% of calories after 17:00. Prior to the start of feeding, we estimated the amount of calories per day required to maintain weight for each participant and this level of calories was kept constant throughout the trial. For 12 weeks, participants were provided all their food and beverages with the exception of water, some non-caloric or low-caloric beverages, and sodium-free spices and herbs. Participants were required to eat one meal at the study center three times per week, except for national holidays; any other exceptions were discussed and approved ahead of time by study staff. On all other days, meals were fully prepared, packed, and labeled for participants to eat at home. Between September 2018 and December 2019, TRIM randomized 41 participants in three cohorts.

### 2.2. Strategies to Promote Adherence

Several strategies were incorporated into TRIM to promote dietary adherence: (1) careful planning of the study schedule and with special menus for holidays, (2) active involvement of dietitians in screening participants, (3) provision of an orientation session and use of a run-in period before randomization to allow participants to understand the study intervention and expectations, (4) allowing participants flexibility to drink some beverages and use some seasonings as described above, and (5) building a trusting relationship between study staff and participants in a friendly dining environment. 

When planning the study schedule, we avoided major holidays as much as possible and conducted no study interventions or visits between the Christmas and New Year’s holidays. When holidays occurred during a feeding period, we provided some accommodations to motivate participants to bring study food with them to their social gatherings. These included special meals and plate decorations and incorporating holiday foods with similar nutrient composition as the main diet on Independence Day and Thanksgiving. 

During screening, a study dietitian met individually with each potential participant to review their food preferences, intolerances, and allergies. Individuals with allergies to frequently-used foods such as milk, egg, peanuts, tree nuts, fish (salmon, tuna, or cod), and wheat, were excluded. Individuals with food intolerances that did not involve an immune response, or who avoided certain foods due to cultural or religious reasons, were included only if we could provide an appropriate substitution. 

Prior to randomization, eligible participants were given an on-site orientation by study dietitians and investigators to review the expectations of the study and answer any participant questions. Then, participants underwent a one-week run-in period where they were given 4 days of the TRF diet and 3 days of the UFP diet. This allowed them to become familiar with the intervention procedures and meal schedules, try most of the study foods, and interact with the research and kitchen staff. Observing the week of run-in also allowed staff members to identify individuals who experienced difficulty adhering to the feeding protocol and exclude them if necessary. During on-site feeding, participants were encouraged to eat in small groups and brainstorm creative ways to consume the provided foods, such as making a smoothie out of frozen fruits and milk. 

Throughout the study, investigators, dietitians, and clinical staff were available to address participant concerns and answer questions. At the participants’ on-site meals during the study period, study dietitians directly observed meal attendance and food consumption and noted any issues. When a deviation occurred, it was addressed between the participant and research staff in a private setting or over the phone. Some deviations could be resolved by providing individual solutions. For example, for participants with difficulty chewing, we sliced their apples or boiled their beef stew longer and chopped the beef more finely when they ate on-site. In all such cases, we aimed to maintain better acceptance of the food without compromising the intervention.

### 2.3. Measuring Adherence

Adherence was measured by participant self-report using a standardized paper form completed by participants for each day during the study period. Using this “daily diary” form (Figure A1 and Figure A2), participants recorded the start time, duration, and the end time of each meal and snack, as well as any uneaten study foods, consumption of non-study foods, consumption of alcoholic and non-alcoholic beverages, and intake of medication and supplements. When participants deviated from the diet (not meal schedule), the specific types of food, quantity, and reason for non-adherence could be reported on the form, but this was not mandated. Daily diary forms were returned to the study dietitians and reviewed for completeness and accuracy on the days participants ate on-site where adherence to meal timing and food consumption was also directly observed by the study dietitians. Participants were encouraged to provide accurate answers and to report any deviations from study protocols in real time to improve information accuracy. Any observed adherence issues during on-site meals were addressed immediately and recorded on the daily diary form either by the participants or the study dietitians with approval from the participants. Due to the nature of the study intervention, this form could not be administered anonymously. We used the daily diary forms to calculate two measures of adherence: (1) adherence to meal timing and (2) adherence to food consumption.

#### 2.3.1. Adherence to Meal Timing

For each meal, adherence to meal timing was classified as perfect, good, or poor. Perfect adherence to meal timing was defined as eating the designated meal entirely within its assigned time window (Table A1) as recorded on the daily diary form. Good adherence was defined as starting or finishing the designated meal up to 30 min outside the time window. Poor adherence was defined as deviating greater than 30 min outside the time window. Good adherence and poor adherence were grouped as non-perfect adherence.

#### 2.3.2. Adherence to Food Consumption

Adherence to food consumption was defined as both eating all study food provided and eating no food from a source outside the study on a designated day as recorded on the daily diary form. Non-adherence to food consumption was defined as not finishing all study food and/or having food from an outside source.

### 2.4. Data Analysis

Adherence measures were calculated as the percentage of participant-days (participants multiplied by number of days in the intervention) meeting adherence criteria. Adherence measures were also calculated on a per-participant basis as the proportion of study days that adherence criteria were met by individual participants. For calculating adherence to meal timing, the lowest level of adherence to any meal in a given day was attributed to the entire day. Daily diary forms from Day 2 of the intervention until final data collection were included in the analysis. Days that participants had an oral glucose tolerance test (OGTT) were excluded because participants were required to be fasting. Days with missing forms were also excluded from analysis because the investigators noted that the reason for missing forms was predominantly not related to adherence (participant forgot to return the form) and the number of missing forms was small.

We examined associations among study arm, day of the week, and week of the study (predictors) and perfect adherence versus non-perfect adherence to meal timing (outcome) using generalized estimating equations (GEE) regression with an exchangeable covariance matrix. The regression models included the predictor of interest as a binary or nominal variable, and a nominal indicator to adjust for study cohort. For models evaluating day of week and week of the study as predictors, analyses were stratified by study arm. We performed the same analyses for the outcome of adherence to food consumption. We considered a *p*-value <0.05 to be statistically significant. All analyses were conducted using STATA software version 14 (StataCorp LP, College Station, TX, USA).

## 3. Results

### 3.1. Participant Characteristics

A total of 41 participants were randomized into the TRF arm (*n* = 21) and the UFP arm (*n* = 20). All participants completed the study. Overall, participants had an average age of 59.4 years, were predominately females (93%), of black race (93%), married at least once (71%), held full-time or part-time employment (71%), and had household incomes greater than 45,000 USD (76%) (Table 1). Participants in the TRF and UFP arms shared similar demographic and socioeconomic characteristics, except for education, where more participants in the TRF than UFP arm received a bachelor’s degree or higher (71% vs. 35%, respectively) (Table 1).

### 3.2. Completeness of Adherence Measurement

There were 3485 total daily diary forms expected (Figure A3). Of these, 3417 (98%) forms were returned (99% returned in TRF, 97% in UFP). Of those returned, 229 forms (6.7%) were excluded from analysis for adherence to meal timing and 147 forms (4.3%) were excluded from analysis for adherence to food consumption based on the criteria noted above in the Data Analysis Section. In total, 3188 forms were analyzed for adherence to meal timing and 3270 were analyzed for adherence to food consumption. 

### 3.3. Adherence to Meal Timing

Data on meal timing are shown in Table 2. There was perfect adherence to meal timing at all meal times on 87% of participant-days, with no significant difference by study arm (TRF, 88% of participant-days and UFP, 85% of participant-days). Good adherence (deviation less than 30 min) occurred on 8% of participant-days in TRF and 10% in UFP, and poor adherence (deviation greater than 30 min) occurred on 4% of participant-days in both TRF and UFP. In terms of each individual meal, there was perfect adherence to meal timing for 93% or greater of all meals in each study arm, with no significant difference by study arm. Among individual participants, achievement of perfect adherence to meal timing ranged from 28% to 100% of study days. The majority of participants (24 of 41, 59%) had perfect adherence to meal timing on 90% of study days or greater (14 in TRF and 10 in UFP) (Table A2). There were nine participants (22%, three participants in TRF and six participants in UFP) who had perfect adherence to meal timing on less than 80% of study days.

As expected, the study intervention resulted in substantial differences in meal timing by treatment arms (Figure 1). Since participants in TRF and UFP shared the same eating windows at breakfast and lunch, meal times were similar in both arms before 13:00. The greatest contrast in meal timing between treatment arms occurred later in the day, which aligned with the intervention design as participants in UFP were required to take a break between 13:00 and 16:00 but could eat until midnight, versus participants in TRF who could eat continuously but were required to finish by 18:00.

### 3.4. Adherence to Food Consumption

The data on food consumption are shown in Table 3. There was adherence to food consumption on 94% of participant-days overall, with no significant difference by study arm. There was no significant difference in reasons for non-adherence to diet by study arm. Most non-adherence events (65% in TRF and 55% in UFP) were due to participants not finishing the study food. A minority of non-adherence events were due to participants eating food outside of the study (22% in TRF and 34% in UFP) or both not finishing the study food and eating outside food on the same day (13% in TRF and 11% in UFP). Among individual participants, adherence to food consumption ranged from 61% to 100% of study days. There were seven participants (17%) who adhered to food consumption on all days of the study period (five participants in TRF and two participants in UFP); there were 26 participants (63%) who adhered to food consumption on 90–99.9% of the study period (12 participants in TRF and 14 participants in UFP); and there were four participants (10%) who adhered to food consumption on less than 80% of study days (three participants in TRF and one participant in UFP) (Table A2).

### 3.5. Differences in Adherence by Day of Week

We examined adherence to meal timing and food consumption by day of the week (Table 4). A significant difference in perfect adherence to meal timing across the week occurred in UFP but not in TRF. In UFP, participants had lower perfect adherence over the weekend (81% of participant-days on Saturday and 78% on Sunday) as compared with on any weekday (all weekdays ≥87%). In TRF, participants had similar perfect adherence to meal timing across the week (all days ≥86%). No significant difference was found in adherence to food consumption across the week in either TRF or UFP.

### 3.6. Differences in Adherence across the Study Period

We also examined adherence to meal timing longitudinally over each week of the 12-week feeding period (Table 5). Significant differences in perfect adherence to meal timing across the feeding period occurred in UFP but not in TRF. In UFP, participants had lower perfect adherence during the first two weeks of intervention (75% of participant-days for Week 1 and 79% for Week 2) as compared with the rest of the weeks (all other weeks ≥81%). In TRF, participants had a similar level of perfect adherence to meal timing across the feeding periods (all weeks ≥83%). No significant difference was found in adherence to food consumption across the entire feeding period in either TRF or UFP.

## 4. Discussion

In this randomized controlled feeding study that compared TRF to UFP, high and similar levels of adherence were achieved in both study arms. Participants achieved perfect adherence to meal timing for 88% of participant-days in TRF and 85% of participant-days in UFP, and completed all study food and ate no outside food for 93% of participant-days in TRF and 95% in UFP. Further, all participants who were randomized completed the study. Therefore, it is feasible to conduct a feeding study enrolling free-living individuals to different eating times and calorie distributions throughout the day with a high degree of adherence to the study intervention.

Other controlled feeding studies have obtained high levels of adherence similar to those obtained in this study, although differences in how adherence was reported make comparisons difficult. Among controlled feeding studies in free-living individuals assessing adherence using participants’ daily report, adherence was reported as the percentage of participant-days [2,17], the number of adherent days per week [18], and averaged number of non-compliance eating episodes [11]. In the Dietary Approaches to Stop Hypertension (DASH) trial, participants adhered on 93–95% of days depending on the study arm [2]. In the OmniHeart trial, participants adhered on 95–96% of days depending on study intervention [17]. In a recent time-restricted feeding study without restricting calories, participants were adherent to their meal schedule on an average of 6.2 days/week in their participants undergoing time-restricted feeding, and the adherence did not change over the 8 weeks of the intervention period [18]. In a recent randomized controlled crossover study on time-restricted feeding, participants had an average of 5.4 episodes of out-of-range eating in the early group versus 2.2 episodes in the late group during the 8 weeks of intervention [11].

There is no consensus on the optimal approach to measuring dietary adherence in feeding studies of free-living individuals [12]. In this study, adherence was assessed by participants’ daily self-reports using a standardized tool, which was an approach that has been widely used in feeding studies involving free-living participants [2,6,7,8,9,11,18,19]. This approach allowed us to collect detailed information on participant adherence, especially to meal timing, which was the focus of this study, while imposing a relatively low burden on participants. However, self-reported measures of adherence rely on participant honesty and accuracy and may be affected by social desirability bias [2,20]. We attempted to minimize inaccurate recall and bias by requiring participants to complete daily diary forms in real time throughout the study, reviewing forms for completeness, and encouraging participants to provide honest answers. Several biological measures have also been used in feeding studies including urinary excretion of electrolytes (sodium, potassium, and phosphorus), urea nitrogen, urine osmolarity, blood levels of nutrients, isotope labeled metabolites, and fecal markers [2,8,9,12,20,21]. While these biological measures can provide objective data on adherence to the study diet, they are unable to assess adherence to meal timing [12].

Published studies presenting adherence to meal timing are sparse. In previous time-restricted feeding studies, adherence was primarily measured by a self-report daily adherence log or survey [6,11,18], a self-report weekly 3-day record [14], staff monitoring on-site and/or remote video by Skype [4,6], frequent staff contacts or dietitian consultation [11,14,15], which are similar to the tools used in TRIM. In a prior controlled feeding study examining the effect of reduced meal frequency, adherence to a single meal per day diet was assessed using blood glucose and triacylglycerol levels collected at three unannounced occasions, which may be an option to provide objective measures of adherence to meal timing in feeding studies, although this adds substantial participant burden [6]. Weigh-backs of containers returned were also used to measure adherence in a recent feeding study [8]; however, this method requires extra staffing and proper handling of returned containers. A promising option for measuring adherence in future feeding studies is the application of smart technology, i.e., adapted ecological momentary assessment via apps or devices such as text messaging or taking meal pictures, which may help minimize recall bias by interacting with participants in real time [22,23].

Participants may face many challenges when participating in a feeding study. In TRIM, this was especially true in the TRF arm in which participants were asked to consume calories on a very different schedule than most U.S. adults. We were concerned that participants randomized to TRF would have lower adherence to dinner because they would find it challenging to finish eating by 18:00. However, the results from our study showed that adherence to both meal timing and food consumption were very similar in the TRF and UFP arms, indicating that the TRF pattern was not burdensome enough to meaningfully affect study adherence. It is possible that participants randomized to TRF found the smaller portion size of dinner to be more tolerable, or that they were highly motivated to adhere to the meal timing, hoping that the diet would yield a weight loss effect. Regarding adherence to food consumption, the findings in this study indicate that eating all study foods was more challenging than refraining from eating foods not part of the study, which has been observed in previous feeding studies [1,2]. These findings highlight the need to motivate and reinforce adherence to meal timing and completion of study foods in both the intervention and control arms of feeding studies examining time-restricted feeding.

We also found subtle differences in adherence by day of the week such that participants randomized to UFP had lower adherence to meal timing on weekends. This may indicate that participants have a different meal schedule for weekends than weekdays, as UFP was designed to represent typical meal timing among general Americans. It is not clear whether offering increased meal flexibility to participants would increase adherence on weekends, and a previous study found worse adherence when a self-selected Saturday meal was allowed [1]. Further, we found lower adherence to the meal timing during the first two weeks of the feeding period in UFP. This may indicate that participants randomized to UFP needed more time to adapt to certain aspects of the meal schedule such as the large dinner size. It is also possible that participants randomized to TRF were highly motivated to adhere to the meal timing hoping that the diet would yield weight loss effect.

Other features of the TRIM study design outside of meal timing may have influenced dietary adherence, including the relatively long duration of feeding, the requirement for specific meal timing, and allowance of weight change during the study. Participants in TRIM ate the same diet for 12 continuous weeks which is more challenging than many other feeding studies which have had a shorter duration and/or breaks in between different diets [2,4,6,7,10,11,15,16]. Results from two post-study surveys of prior participants in feeding studies ranging from 6 to 24 weeks duration found that dietary adherence was highest in the 12-week study and there was overall no association between study length and adherence, suggesting that factors beyond study length play an important role [1,13]. TRIM is also unusual among feeding studies which typically monitor participants’ weights during the study period and adjust their diet calories to maintain a stable weight [2,4,5,6]. In TRIM, weight change was the primary outcome, therefore, participants maintained the same calorie level from baseline, and weight was not monitored during the study except to ascertain outcomes. However, this required complete reliance on the baseline estimation of caloric requirements, and participants could not be given their weight during the study period despite several participants expressing interest in knowing this.

This study has limitations. First, all self-reported nutrition assessment tools may be affected by recall bias and social desirability bias. We attempted to minimize bias by encouraging participants to complete daily diary forms in real time throughout the study and encouraging accurate reporting. Second, this was a relatively small study with 41 participants who were predominantly non-Hispanic black race/ethnicity and female, therefore, the results may not be generalizable to all populations. Notwithstanding these limitations, this study used multiple methods to promote adherence, which provided a basis for optimizing adherence in future feeding studies involving a meal timing component. Overall, this study demonstrated that a controlled feeding study randomizing free living participants to different meal timings could be conducted with complete follow-up and high adherence to meal timing and study diet. Participants in the TRF arm achieved a similar, high level of adherence as in the control arm, indicating that randomization to a time-restricted meal schedule is well-tolerated and feasible.

## Figures and Tables

**Figure 1 nutrients-14-02283-f001:**
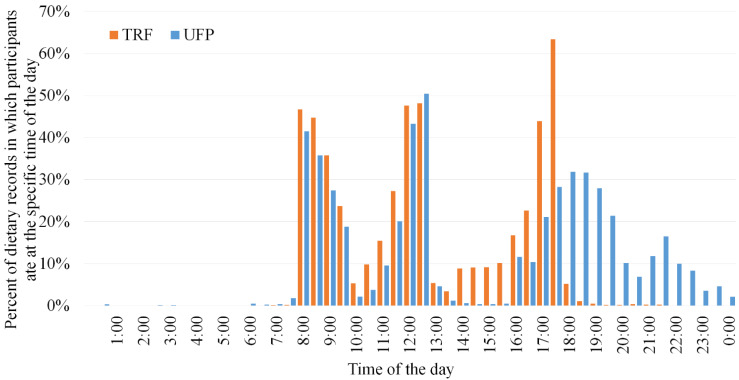
Distribution of meal timing during the study period by study arm. Abbreviations: TRF, time-restricted feeding arm; UFP, usual feeding pattern arm.

**Table 1 nutrients-14-02283-t001:** Participant baseline characteristics overall and by study arm.

Characteristic	All Participants Mean (SD) or Number (%)	TRF ^1^ Mean (SD) or Number (%)	UFP ^2^ Mean (SD) or Number (%)
Age, years	59.4 (7.1)	59.7 (7.0)	59.1 (7.5)
**Age categories**			
40–49 years	4 (9.8)	2 (9.5)	2 (10.0)
50–59 years	13 (31.7)	6 (28.6)	7 (35.0)
60–69 years	24 (58.5)	13 (61.9)	11 (55.0)
**Sex**			
Female	38 (92.7)	19 (90.5)	19 (95.0)
Male	3 (7.3)	2 (9.5)	1 (5.0)
**Race**			
Black	38 (92.7)	20 (95.2)	18 (90.0)
White	3 (7.3)	1 (4.8)	2 (10.0)
**Marital status**			
Single	12 (29.3)	5 (23.8)	7 (35.0)
Married	18 (43.9)	8 (38.1)	10 (50.0)
Widowed	3 (7.3)	1 (4.76)	2 (10.0)
Divorced\Separated	8 (19.5)	7 (33.3)	1 (5.0)
**Education**			
<Bachelor	19 (46.3)	6 (28.6)	13 (65.0)
Bachelor	12 (29.3)	9 (42.9)	3 (15.0)
>Bachelor	10 (24.4)	6 (28.6)	4 (20.0)
**Employment status**			
Full-time	24 (58.5)	14 (66.7)	10 (50.0)
Part-time	5 (12.2)	2 (9.5)	3 (15.0)
None	12 (29.3)	5 (23.8)	7 (35.0)
**Yearly household income ^3^**			
<$45,000	9 (23.7)	4 (20.0)	5 (27.8)
$45,000–<$75,000	14 (36.8)	7 (35.0)	7 (38.9)
>$75,000	15 (39.5)	9 (45.0)	6 (33.3)

^1^ TRF, time-restricted feeding arm; ^2^ UFP, usual feeding pattern arm; ^3^ income missing for 1 participant in the TRF arm and for 2 participants in the UFP arm.

**Table 2 nutrients-14-02283-t002:** Adherence to meal timing overall and by study arm.

Number (%) of Participant-Days				
All Participants		TRF ^1^	UFP ^2^	*p*-Value ^3^
**Adherence to all meal**	3188	1649	1539	0.36
**times**				
Perfect adherence	2763 (87)	1448 (88)	1315 (85)	
Good adherence	291 (9)	135 (8)	156 (10)	
Poor adherence	134 (4)	66 (4)	68 (4)	
**Breakfast**	3268	1693	1575	0.14
Perfect adherence	3170 (97)	1654 (98)	1516 (96)	
Good adherence	61 (2)	24 (1)	37 (2)	
Poor adherence	37 (1)	15 (1)	22 (1)	
**Lunch**	3272	1692	1580	0.75
Perfect adherence	3059 (93)	1582 (94)	1477 (93)	
Good adherence	123 (4)	54 (3)	69 (4)	
Poor adherence	90 (3)	56 (3)	34 (2)	
**Dinner**	3254	1678	1576	0.10
Perfect adherence	3123 (96)	1630 (97)	1493 (95)	
Good adherence	83 (3)	22 (1)	61 (4)	
Poor adherence	48 (1)	26 (2)	22 (1)	
**Snack**	3219	1665	1554	0.10
Perfect adherence	3053 (95)	1572 (94)	1481 (95)	
Good adherence	109 (3)	69 (4)	40 (3)	
Poor adherence	57 (2)	24 (1)	33 (2)	

^1^ TRF, time-restricted feeding arm; ^2^ UFP, usual feeding pattern arm; ^3^
*p*-value compares perfect adherence and non-perfect adherence in TRF vs. UFP using GEE regression adjusted for study cohort; non-perfect adherence includes good adherence and poor adherence; good adherence defined as 1–30 min deviations from study meal schedule; poor adherence defined as >30 min deviations from study meal schedule.

**Table 3 nutrients-14-02283-t003:** Adherence to food consumption overall and by study arm.

Number (%) of Participant-Days				
	All Participants	TRF ^1^	UFP ^2^	*p*-Value ^3^
**Adherence to study diet**	3270	1691	1579	
Adherence	3063 (94)	1570 (93)	1493 (95)	0.65
Non-adherence				
Had food left-over only	126 (4)	79 (5)	47 (3)	0.39
Ate outside food only	56 (2)	27 (2)	29 (2)	0.50
Had left-over and ate outside food	25 (1)	15 (1)	10 (1)	0.80

^1^ TRF, time-restricted feeding arm; ^2^ UFP, usual feeding pattern arm; ^3^
*p*-value compares TRF vs. UFP using GEE regression adjusted for study cohort.

**Table 4 nutrients-14-02283-t004:** Distribution of perfect adherence to meal timing and adherence to food consumption by day of week.

	Perfect Adherence to Meal Timing, % of Participant-Days				Adherence to Food Consumption, % of Participant-Days			
	TRF ^1^	*p*-Value ^3^	UFP ^2^	*p*-Value ^3^	TRF ^1^	*p*-Value ^3^	UFP ^2^	*p*-Value ^3^
Monday	87	0.72	89	0.002	91	0.05	93	0.89
Tuesday	88		87		95		94	
Wednesday	87		89		95		96	
Thursday	90		87		93		94	
Friday	88		87		95		94	
Saturday	88		81		94		94	
Sunday	86		78		89		96	

^1^ TRF, time-restricted feeding arm; ^2^ UFP, usual feeding pattern arm; ^3^
*p*-value is for difference in adherence across day of the week by GEE stratified by study arm, adjusting for cohort.

**Table 5 nutrients-14-02283-t005:** Distribution of perfect adherence to meal timing and adherence to food consumption by study week over the 12-week feeding period.

	Perfect Adherence to Meal Timing, % of Participant-Days				Adherence to Food Consumption, % of Participant-Days			
Week	TRF ^1^	*p*-Value ^3^	UFP ^2^	*p*-Value ^3^	TRF ^1^	*p*-Value ^3^	UFP ^2^	*p*-Value ^3^
1	85	0.39	75	0.03	90	0.15	92	0.42
2	84		79		93		94	
3	89		90		93		92	
4	83		84		93		98	
5	86		91		95		97	
6	89		85		95		96	
7	91		84		86		94	
8	90		90		94		98	
9	87		81		92		95	
10	90		87		93		93	
11	92		88		95		94	
12	88		92		96		93	

^1^ TRF, time-restricted feeding; ^2^ UFP, usual feeding pattern arm; ^3^
*p*-value is for difference in adherence across week of the study period by GEE stratified by study arm, adjusting for cohort.

## Data Availability

The data presented in this study are available on request from the corresponding author. The data are not publicly available due to privacy reason.

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
