# Peer review of "Adherence to Diet and Meal Timing in a Randomized Controlled Feeding Study of Time-Restricted Feeding"

_nutrients, 2022, doi:10.3390/nu14112283_

Round 1
Reviewer 1 Report
Thanks to the editor for the invitation. In this study, the authors have investigated the adherence to diet and meal Timing in a randomized controlled feeding study of Time-restricted Feeding. This is well-designed and promising trial, whereas I did not fully get the aim of the current manuscript. If I understand correctly, the purpose of this study is to prepare methodologically for subsequent real findings. I don’t have major concern on this manuscript and look forward to the publication of subsequent results.
One question: For the self-reported questionnaire (2.3 in method section), are there any monitoring measures to verify the accuracy of the questionnaire?
Author Response
Response to Reviewer 1 Comments
- For the self-reported questionnaire (2.3 in method section), are there any monitoring measures to verify the accuracy of the questionnaire?
Response 1: Thank you for this comment. In addition to daily self-report of adherence, participants were also directly observed by study dietitians for meal attendance and food consumption 3 times per week at which time and food adherence issues were reviewed and daily diary forms were checked for completeness and accuracy. The process was also described briefly in 2.2 Strategies to Promote Adherence. Although we used direct observation for monitoring adherence for on-site meals, any observed adherence issues would be recorded on the daily diary form either by the participant or the study dietitians with approval from the study participant. Therefore, for analytical purpose, we only used data from daily diary forms to assess adherence. In a prior study (PMID: 16182647) with independent measure of adherence such as a post-study anonymous survey indicated good adherence but is subject to substantial recall bias. In our study, we encouraged participants to fill out the questionnaire simultaneously as they ate to minimize this issue and improve accuracy. In prior feeding studies (PMID: 10450298) self-reported daily diaries were the most sensitive method for detecting non-adherence, which is why they were used in this study. We have revised the 2.3 Measuring Adherence to add these details.
Reviewer 2 Report
I have carefully studied the manuscript sent and I can see that your team had a laborious activity, with a well-defined purpose. The methods used to conduct the study were correct and the statistical tests were applied accordingly.
Unfortunately, after such intense and documented work, I find that the study has some limitations recognized even by the authors. These limitations actually cancel all work done by the team.
That is why I recommend that the number of patients included in the study be higher and that the study allow the elaboration of conclusions not only of a rather ambiguous and poor demonstration of the work that was submitted for this study.
I look forward to an improved version of this study that will allow for clearer and more effective conclusions.
Author Response
Response to Reviewer 2 Comments
Point 1: recommend that the number of patients included in the study be higher
Response 1: Thank you for your review. The study sample size was determined by the power calculations for the primary outcome of weight change, and we exceeded our recruitment targets. We believe that our sample size is sufficient to achieve the objectives of this study, which are to describe study adherence and our approach to assessing and maximizing adherence. In fact, a strength of this study is that it is one of the largest feeding studies examining time restricted feeding that has ever been performed. We therefore do not feel that sample size is a significant limitation. While our study was a majority black race and female, which may affect interpretation of generalizability, an important aspect of our study design is how we tailored our procedures with input from our participants to achieve good adherence in the population being recruited. As this challenge is common to all feeding studies, we believe our study provides a valuable contribution and roadmap for how to achieve successful adherence for nutrition research in a diverse range of populations.
Reviewer 3 Report
Dear Authors,
I had the opportunity to revise your manuscript and I think it does not need any correction. It is rare to have the pleasure to read such a high-quality manuscript capable to capture the reader's attention. Best of luck and Kind regards.
Author Response
Thank you for noting the hard work we put into the study and your kind comments. Again, thank you for taking the time to review our manuscript and provide such positive feedback.